# Measuring of Advanced Glycation End Products in Acute Stroke Care: Skin Autofluorescence as a Predictor of Ischemic Stroke Outcome in Patients with Diabetes Mellitus

**DOI:** 10.3390/jcm11061625

**Published:** 2022-03-15

**Authors:** Alexandra Filipov, Heike Fuchshuber, Josephine Kraus, Anne D. Ebert, Vesile Sandikci, Angelika Alonso

**Affiliations:** Department of Neurology, Medical Faculty Mannheim, University of Heidelberg, 68167 Mannheim, Germany; heike_fuchshuber@web.de (H.F.); josephine.kraus@umm.de (J.K.); anne.ebert@umm.de (A.D.E.); vesile.sandikci@umm.de (V.S.); angelika.alonso@umm.de (A.A.)

**Keywords:** stroke outcome, diabetes mellitus, hyperglycemia, skin autofluorescence, advanced glycation end products, poststroke complications

## Abstract

*Background:* Patients with diabetes mellitus (DM) are known to show poor recovery after stroke. This specific burden might be due to acute and chronic hyperglycemic effects. Meanwhile, the underlying mechanisms are a cause of discussion, and the best measure to predict the outcome is unclear. Skin autofluorescence (SAF) reflects the in-patient load of so-called advanced glycation end products (AGEs) beyond HbA1c and represents a valid and quickly accessible marker of chronic hyperglycemia. We investigated the predictive potential of SAF in comparison to HbA1c and acute hyperglycemia on the functional outcome at 90 days after ischemic stroke in a cohort of patients with DM. *Methods:* We prospectively included 113 patients with DM type 2 hospitalized for acute ischemic stroke. SAF was measured on each patient’s forearm by a mobile AGE-Reader mu© in arbitrary units. HbA1c and the area under the curve (AUC) of the blood sugar profile after admission were assessed. Functional outcome was assessed via phone interview after 90 days. A poor outcome was defined as a deterioration to a modified Rankin Scale score ≥ 3. A good outcome was defined as a modified Rankin Scale score < 3 or as no deterioration from premorbid level. *Results:* Patients with a poor outcome presented with higher values of SAF (mean 3.38 (SD 0.55)) than patients with a good outcome (mean 3.13 (SD 0.61), *p* = 0.023), but did not differ in HbA1c and acute glycemia. In logistic regression analysis, age (*p* = 0.021, OR 1.24 [1.12–1.37]) and SAF (*p* = 0.021, OR 2.74 [1.16–6.46]) significantly predicted a poor outcome, whereas HbA1c and acute glycemia did not. Patients with a poor 90-day outcome and higher SAF experienced more infections (4.2% vs. 33.3% (*p* < 0.01)) and other various in-hospital complications (21.0% vs. 66.7% (*p* < 0.01)) than patients with a good outcome and lower SAF levels. *Conclusions:* SAF offers an insight into glycemic memory and appears to be a significant predictor of poor stroke outcomes in patients with DM exceeding HbA1c and acute glycemia. Measuring SAF could be useful to identify specifically vulnerable patients at high risk of complications and poor outcomes.

## 1. Introduction

Around 30% of patients in ischemic stroke care suffer from diabetes mellitus (DM). Concomitantly, due to acute and chronic hyperglycemic effects, patients with DM show poor recovery after stroke [1]. HbA1c from nonenzymatic glycation of hemoglobin represents the best-established marker of chronic hyperglycemia regarding the last three months. Meanwhile, different long-lasting molecules underlie similar transformations and form the group of advanced glycation end products (AGEs), also known as glycemic memory [2]. Skin autofluorescence (SAF) represents a valid, quick and noninvasive approach to measure AGEs in vivo [3] and is a marker of vasculopathy in DM type 2 [4]. We aimed to investigate the predictive potential of SAF as a surrogate of long-term hyperglycemia in comparison to HbA1c as marker of intermediate glycemia and acute hyperglycemia on stroke outcome in a cohort of patients with DM.

## 2. Materials and Methods

From December 2018 to September 2020, patients were prospectively recruited at the University Hospital of Mannheim, Germany. Our assessments were based on the most prevalent scoring scales in stroke medicine [5]. The modified Rankin scale (mRS) is a 7-item scale indicating functional dependency. A score of 0 is considered no disability, 5 is disability requiring constant care for all needs and 6 is death. A score of more than 2 is the hallmark of functional dependency. The Barthel Index (BI) is a scale used to measure performance in activities of daily living according to 10 different variables. The National Institutes of Health Stroke Scale (NIHSS) is a 15-item neurologic examination scale evaluating the effect of cerebral infarction on the levels of consciousness, language, neglect, visual field, extraocular movement, motor strength, ataxia, dysarthria and sensory loss. We included adult patients with known DM type 2 or HbA1c ≥ 6.5% at admission hospitalized for ischemic stroke (according to World Health Organisation definition [6]) presenting within 3 days after symptom onset with a persistent deficit ((mRS) score ≥ 1). Written consent was obtained from the patient or their legal representative. Patients necessitating hemodialysis were excluded [7]. SAF was measured bedside on the patient’s volar forearm by a mobile AGE-Reader mu© (DiagnOptics Technologies B.V., Groningen, The Netherlands). According to usage instructions, the patient placed their volar forearm on the measurement window where light was radiated on the previously degreased skin. The reflected light was registered to measure SAF that was displayed within 12 s in arbitrary units (AU) (for validation study and technical details, see Meerwaldt et al., 2004 and 2005 [8,9]). Three measurements were performed bedside with a slight change in the forearm’s position. The mean value was calculated for further analysis as intraindividual variance in same-day measurement ranges around 5% according to reference data [8] without relevant postprandial changes [10]. A routine blood analysis included HbA1c. From routine capillary blood sugar profiling, we calculated the area under the curve (AUC) in mg/mL × 24 h, representing acute glycemia with respect to the first two days after admission, standardized in 24 h. Insulin was administered after blood sugar measuring, as clinically required. Baseline parameters from medical history including preexisting functional deficit (pre-mRS) were registered, as well as severity of stroke by NIHSS. If indicated, acute revascularization therapy was performed according to local standards. We recorded in-hospital complications such as (symptomatic) intracranial hemorrhage ((S) ICH) [11] in follow-up cranial imaging, as well as infectious complications [12]. Other complications (recurrent stroke, epileptic seizures, delirium, acute renal failure, thrombosis, pulmonary embolism, myocardial infarction and others) were recorded if they required diagnostic or therapeutic measures. For follow-up, we performed a phone interview after 90 (±3) days poststroke and determined mRS and BI. A poor functional outcome whilst taking into account prior deficit was defined as a deterioration from premorbid mRS to mRS ≥ 3 at 90 days poststroke. A good outcome was defined as a mRS < 3 or as no deterioration from premorbid mRS.

Statistical analysis was performed with SPSS^®^ 27.0 (IBM, Armonk, New York, NY, USA). *p* values < 0.05 were considered statistically significant. We compared baseline and clinical characteristics, in-hospital complications and 90 days of BI between patients with a poor and a good 90-day outcome. Intergroup differences were assessed using *t*-test for metric variables, Mann–Whitney U test for ordinal variables and Chi^2^ test/Fisher’s exact test for categorical variables as appropriate. We further performed a multiple logistic regression analysis, including the preliminarily defined predictors SAF, HbA1c and AUC as glycemic variables adjusted for age and NIHSS at admission as the strongest known predictors of a poor 90-day outcome [13].

## 3. Results

A total of 113 patients (mean age 71.4 years, SD 10.29; 59.3% male) were included. There was no significant correlation either between SAF and HbA1c (Pearson’s correlation coefficient, r = 0.02) or between SAF and age (r = 0.17). Furthermore, we did not find a correlation between NIHSS at admission and either glucose at admission (Spearman’s rank correlation coefficient, ρ = 0.041) or glycemic AUC (ρ = 0.029). After three months, we were unable to follow up on six patients (5.3%). The premorbid deficit was low in our cohort: before the index stroke, 86.7% of the patients were functionally independent, as indicated by mRS ≤ 2. On day 90, this was the case for only 52.3% (see Figure 1). Additionally, 90 days poststroke, 62 (57.9%) patients showed a good outcome, while 45 (42.1%) showed a poor outcome according to our definition.

When comparing patients with good versus poor outcome, (see in Table 1) patients with poor outcomes were older (mean age 69.0 years (SD 9.57) vs. 76.3 years (SD 9.10), *p* < 0.001) and had a higher level of premorbid functional deficit (pre-mRS: median 0 (IQR 0; 0) vs. 1 (IQR 0; 3), *p* < 0.001; pre-BI: median 100 (IQR 100; 100) vs. 100 (IQR 85; 100), *p* < 0.001). Male patients were more likely to achieve a good outcome (72.6% vs. 40.0%, *p* < 0.001). Patients with a poor outcome exhibited more often known macrovascular disease (32.3% vs. 60.0%, *p* = 0.004) and renal failure (29.0% vs. 48.9%, *p* = 0.036) and were more often under antithrombotic treatment (27.4% vs. 48.9%, *p* = 0.023). Instead, patients with a good outcome were more often under a combination of basal insulin and oral antidiabetic treatment (BOT) (27.4% vs. 4.4%, *p* = 0.002). Considering stroke characteristics, patients with a good outcome showed more frequently infratententorial strokes (30.6% vs. 11.1%, *p* = 0.017). There was no difference considering stroke outcome and stroke etiology in our cohort.

Considering the severity of stroke, patients with a poor outcome showed higher NIHSS scores at admission (median 10 (IQR 5; 16) vs. median 4 (IQR 2; 6), *p* < 0.001), and they received more frequently revascularization therapy (55.6% vs. 33.9%, *p* = 0.025). There was no significant group difference concerning intravenous thrombolysis, but a higher frequency of mechanical thrombectomy in patients with poor outcome (24.4% vs. 9.7%, *p* = 0.039). Complications during the hospital stay did not differ between patients with poor outcome and good outcome in terms of hemorrhagic complications, whereas the rate of intracerebral hemorrhage was generally low in our sample. Poststroke infection occurred more often in patients with poor outcome (33.3% vs. 4.8%, *p* < 0.001) as well as other complications during hospital care (66.7% vs. 21.0%, *p* < 0.001). The total in-patient mortality rate amounted to 3.5%. Among patients with a poor outcome, 8.9% died during the initial hospital stay.

Patients with a poor versus good outcome did not differ in admission glucose, in glycemic AUC, or in HbA1c. However, patients with a poor outcome showed higher SAF (mean 3.13 (SD 0.61) vs. mean 3.38 (SD 0.55), *p* = 0.023) (see Figure 2). 

Logistic regression analysis revealed rising age (*p* = 0.021; odds ratio (OR) 1.07 [1.01–1.12]) and rising NIHSS at admission (*p* < 0.001, OR 1.24 [1.12–1.37]) as predictors being significantly associated with a poor outcome. Regarding glycemic variables, rising SAF turned out to be significantly associated with a poor outcome (*p* = 0.021, OR 2.74 [1.16–6.46]). Meanwhile, HbA1c and AUC did not add significant prediction to the model (see in Table 2).

## 4. Discussion

The mechanisms mediating poor stroke outcome in patients with DM might consist of acute and chronic hyperglycemic effects, although the best measure of hyperglycemia to predict outcome is largely unknown [1].

### 4.1. Troubled Water: Acute Hyperglycemia

Patients with DM are specifically prone to stress hyperglycemia in the context of a severe illness such as stroke [14]. Acute hyperglycemia has been associated with poor stroke outcome, as it was supposed to drive ischemic damage [15]. On the other hand, interventions with aggressive insulin therapy in acute stroke care were not beneficial [16,17]. So, given a connection between hyperglycemia and poor outcome, cause and effect are not clearly attributable. Most prior studies investigating the impact of acute hyperglycemia on stroke outcome have referred to admission glucose and used different arbitrary cut-off values to define hyperglycemia [18]. In this regard, Fuentes et al., (2009) performed blood sugar profiling for 48 h postadmission and confirmed hyperglycemia exceeding 155 mg/dL to be a significant predictor of a poor outcome. In our study, we did not focus on a cut-off value, as we expected expansive glycemic variations in our cohort. In an attempt to meet and objectify the glycemic ups and downs as a dynamic value, we operationalized acute glycemia as the AUC of the blood sugar profile postadmission. Interestingly, patients with a poor and a good outcome did not differ in acute glycemia, neither in admission glucose nor in glycemic AUC. Additionally, AUC was not significantly associated with a poor outcome in logistic regression analysis. In our cohort, neither admission glucose nor AUC correlated with the NIHSS at admission. Accordingly, our data do not support the theory of hyperglycemic derailment in the context of severe stroke in patients with DM. It must be considered that revascularization therapy can result in a reversal of initially severe stroke symptoms. Nevertheless, in our cohort, patients with a poor outcome more frequently underwent acute therapy and thrombectomy, implying only moderate success. On the other hand, in lacunar stroke, mild hyperglycemia might be even favorable [19]. However, according to our results, we cannot attribute a poor stroke outcome to acute hyperglycemia. 

### 4.2. The Foot of the Iceberg: Chronic Hyperglycemia

Meanwhile pre-stroke glycemic control might predict stroke outcome [20,21,22,23]. In our study, patients with good and poor outcomes differed only in SAF regarding glycemic variables, and SAF was the only glycemic predictor significantly associated with a poor outcome, even when adjusting for age and NIHSS. An increase in SAF in one AU was associated with an approximately three-fold risk of a poor outcome on day 90 (OR 2.74). The SAF values we measured lay slightly above the range of age-adapted reference values for patients with DM [4], reflecting the specific vascular risk in our cohort of acute stroke patients. We deduce that SAF reflecting long-term glycemic control is supposed to have a higher impact on stroke outcome than HbA1c or acute glycemia. Possible mechanisms by which chronic hyperglycemia affects stroke outcome include preexisting vascular damage on the macro- and microvascular level impairing collateral flow. Regarding the molecular level, accumulated AGEs are supposed to mediate a self-perpetuating chronic vascular inflammation [24], mainly by interaction with their receptor RAGE (receptor for advanced glycation end products), leading to endothelial dysfunction and arterial stiffness [25], hypercoagulation, diminished fibrinolysis and vasoconstriction [26]. An excess of AGE-RAGE interaction-related downstream inflammatory markers is likely to increase poststroke inflammation, which is known to increase ischemic damage within the brain but also leads to systemic effects such as cardiac injury [27]. This effect seems to be most important in cardioembolic stroke, which was the most frequent subtype in our sample without having a statistical effect on outcome, likely due to a limited sample size. Additionally, AGE-RAGE-mediated effects may promote ICH by blood–brain-barrier disruptions [28] and may increase susceptibility to infectious complications [29]. In our sample, patients with a poor outcome and with higher SAF levels showed more infectious [30] and other in-hospital complications, which are known to impair long-term outcome poststroke [13] on a sensorimotor but also on a cognitive level, especially when combined with renal failure [31]. It seems reasonable that patients with a good outcome and lower SAF benefitted from a better long-term metabolic control prior to the index stroke. Our cohort reflects this point, as patients with a good outcome were more often under BOT, implying a more sophisticated antidiabetic treatment.

We can assume that SAF offers an insight to the extent of the diabetic burden being predictive for stroke outcome in DM, and HbA1c remains the “tip of the iceberg”.

## 5. Limitations

This was a monocentric study in a local urban population, and a certain selection bias concerning standards of acute stroke treatment and further rehabilitation can be expected. The limited number of included patients a priori impeded an exhaustive prediction model with respect to additional potential predictors. The follow-up interviewer was not blinded for glycemic values, allowing a certain rater bias. The measuring of acute glycemia was not continuous but based on blood sugar profile. Still, we found SAF to have the highest predictive value on stroke outcome amongst glycemic variables when controlling for age and severity of stroke. An unexpected finding from our cohort was an important sex-dependent difference in stroke outcome. A possible explanation could be higher age and higher premorbid dependency in female patients [32].

## 6. Conclusions and Future Perspectives

According to our results, SAF, representing long-term glycemic memory, is a significant predictor of a poor functional outcome after ischemic stroke in patients with DM and exceeds HbA1c and acute hyperglycemia in its predictive value. SAF might be a useful tool to identify patients at high risk of complications and poor outcome requiring special attention (for example, preventive antibiotics, prolonged monitoring, adapted antithrombotic treatment). Our study must be considered preliminary. Larger neurovascular patient populations need to be investigated for SAF in the form of registries to create a more exhaustive prediction model and to establish a sensitive and specific cut-off value to distinguish patients at high risk of a poor outcome.

Regarding potential specific therapeutic interventions in the context of acute stroke, it might not be possible to reverse the weight of an iceberg that has accumulated over the years. However, to remain with the allegory, investigating water for potentially assailable key point biomarkers along the RAGE axis could offer future opportunities. For example, soluble RAGE showed a promising ability to counterbalance endothelial dysfunction in a mouse model in the short term [33]. Along these lines, future research is needed.

## Figures and Tables

**Figure 1 jcm-11-01625-f001:**
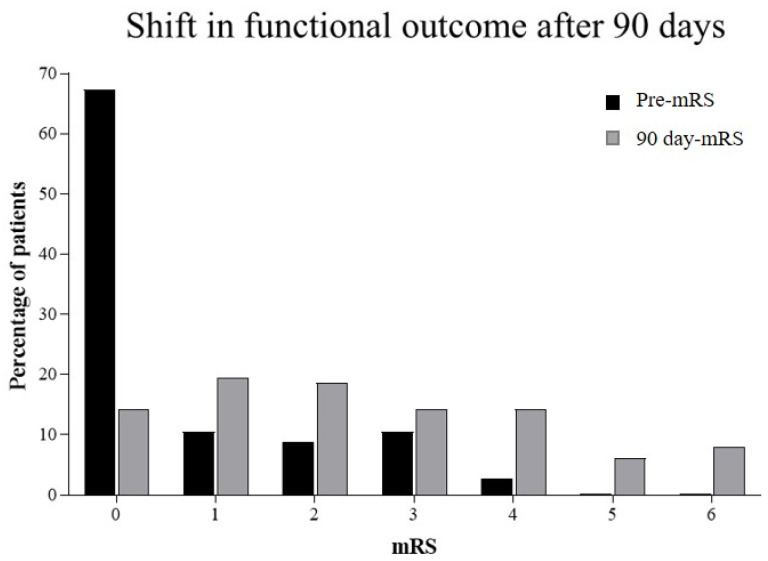
Shift in functional outcome after 90 days: premorbid modified Rankin scale (Pre-mRS; *n* = 113), modified Rankin Scale on day 90 (90 d-mRS; *n* = 107).

**Figure 2 jcm-11-01625-f002:**
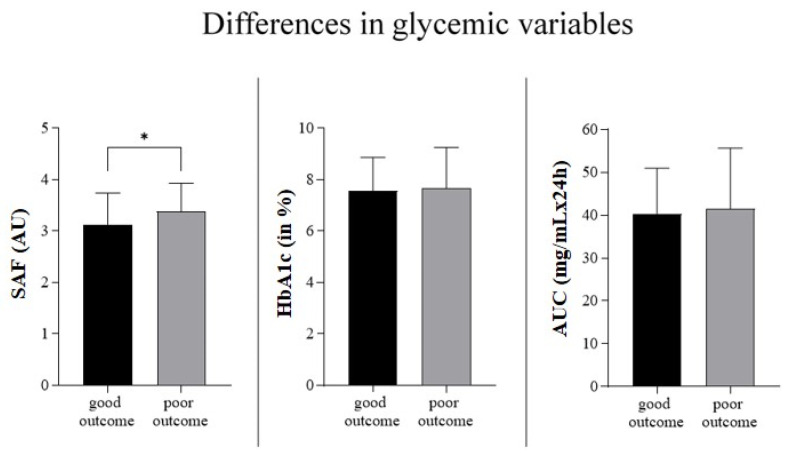
Differences in glycemic variables according to 90-day outcome: mean and standard deviation of SAF, HbA1c and AUC. * Significant difference, skin autofluorescence (SAF), area under the curve (AUC), arbitrary units (AU).

**Table 1 jcm-11-01625-t001:** Baseline characteristics.

Population	Good Outcome (90 d mRS < 3 or No Deterioration)	Poor Outcome (90 d mRs ≥ 3 and Deterioration)	*p*
*n*	62	45	
Age, mean (sd) [years]	69.0 (9.57)	76.3 (9.10)	<0.001 *
Male, *n* (%)	45 (72.6)	18 (40.0)	0.001 *
Premorbid-mRS, median (IQR)	0 (0; 0)	1 (0; 3)	<0.001 *
Premorbid-BI, median (IQR)	100(100; 100)	100 (85; 100)	<0.001 *
Risk factors			
Hypertension, *n* (%)	54 (87.1)	40 (88.9)	0.779
Hyperlipidemia, *n* (%)	20 (32.3)	21 (46.7)	0.130
Atrial fibrillation, *n* (%)	13 (21.0)	17 (37.8)	0.056
Macrovascular disease, *n* (%)	20 (32.3)	27 (60.0)	0.004 *
Renal failure, *n* (%)	18 (29.0)	22 (48.9)	0.036 *
Previous stroke, *n* (%)	10 (16.1)	7 (15.6)	0.936
Smoking, *n* (%)	11 (17.7)	4 (8.9)	0.263
Alcohol abuse, *n* (%)	3 (4.8)	1 (2.2)	0.637
Premedication			
Oral anticoagulation, *n* (%)	10 (16.1)	7 (15.6)	0.936
Antithrombotic agent, *n* (%)	17 (27.4)	22 (48.9)	0.023 *
Statin, *n* (%)	28 (45.2)	27 (60.0)	0.130
Antihypertensive medication, *n* (%)	46 (74.2)	39 (86.7)	0.115
BOT, *n* (%)	17 (27.4)	2 (4.4)	0.002 *
Insulin, *n* (%)	22 (35.5)	14 (31.1)	0.637
Oral antidiabetic, *n* (%)	45 (72.6)	25 (55.6)	0.068
Glycemia			
SAF, mean (sd) [AU]	3.13 (0.61)	3.38 (0.55)	0.023 *
AUC, mean (sd) [mg/(mL × 24 h)]	40.38 (10.58)	41.49 (14.16)	0.647
HbA1c, mean (sd) [%]	7.57 (1.29)	7.67 (1.58)	0.718
Admission variables			
NIHSS, median (IQR)	4 (2; 6)	10 (5; 16)	<0.001 *
Systolic blood pressure, mean (sd) [mmHg]	170.51 (32.35)	163.81 (24.48)	0.285
Plasma glucose, mean (sd) [mg/dL]	191.2 (65.01)	197.84 (79.48)	0.637
Acute revasculating therapy, *n* (%)	21 (33.9)	25 (55.6)	0.025 *
Intravenous thrombolysis, *n* (%)	19 (30.6)	20 (44.4)	0.143
Mechanical thrombectomy, *n* (%)	6 (9.7)	11 (24.4)	0.039
Complications in stay			
ICH, *n* (%)	13 (21.0)	11 (24.4)	0.670
SICH, *n* (%)	0 (0.0)	1 (2.2)	0.421
Poststroke infection, *n* (%)	3 (4.8)	15 (33.3)	<0.001 *
Death, *n* (%)	0 (0.0)	4 (8.9)	0.029 *
Other complications, *n* (%)	13 (21.0)	30 (66.7)	<0.001 *
90 d Outcome			
90 d mRS, median (IQR)	1 (0; 2)	4 (3; 5)	<0.001 *
90 d Barthel, median (IQR)	100 (100; 100)	35 (0; 65)	<0.001 *
Stroke characteristics			
Supratentorial, *n* (%)	48 (77.4)	40 (88.9)	0.125
Infratentorial, *n* (%)	19 (30.6)	5 (11.1)	0.017
Supratent. and Infratent., *n* (%)	6 (9.7)	0 (0.0)	0.039
Large artery disease, *n* (%)	6 (9.7)	9 (20.0)	0.129
Small artery disease, *n* (%)	15 (24.2)	10 (22.2)	0.812
Proximal embolism, *n* (%)	41 (66.1)	28 (62.2)	0.677

*p*-values < 0.005 are considered statistically significant; * significant, (%) percentage of outcome quality, day (d), number (*n*), skin autofluorescence (SAF), arbitrary unit (AU), basal insulin and oral antidiabetic treatment (BOT), area under the curve (AUC), modified Rankin Scale (mRS), National Institutes of Health Stroke Scale (NIHSS), Barthel Index (BI), intracerebral hemorrhage (ICH), symptomatic intracerebral hemorrhage (SICH), standard deviation (SD), interquartile range (IQR).

**Table 2 jcm-11-01625-t002:** Predictors of outcome.

Predictor	*p*	OR [CI]
Age [years]	0.021 *	1.07 [1.01–1.12]
NIHSS [/]	<0.001 *	1.24 [1.12–1.37]
HbA1c [%]	0.520	-
AUC [mg/mL × 24 h]	0.397	-
SAF [AU]	0.021 *	2.74 [1.16–6.46]

*p*-values < 0.05 are considered statistically significant; * significant; OR: odds ratio; CI: confidence interval, skin autofluorescence (SAF), area under the curve (AUC), National Institutes of Health Stroke Scale (NIHSS).

## Data Availability

The data that support the findings of this study are available on reasonable request from the corresponding author. The data are not publicly available due to containing information that could compromise the privacy of research participants.

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
