# Peer review of "Measuring of Advanced Glycation End Products in Acute Stroke Care: Skin Autofluorescence as a Predictor of Ischemic Stroke Outcome in Patients with Diabetes Mellitus"

_jcm, 2022, doi:10.3390/jcm11061625_

Round 1

Reviewer 1 Report

This article investigated whether skin autofluorescence which reflects the load of advanced glycation can be a predictor of ischemic stroke outcome in diabetes patients. The idea is interesting, and the design and writing are good. However, I have some suggestions to improve the manuscript.

  • The authors should give the details of how to calculate Rankin Scale score in methods.
  • What’s the Barthel Index (BI)? How to calculate it? Authors should also give more information in the methods.
  • In the multiple logistic regression analysis, other risk factors such as hypertension, hyperlipidemia, smoking and alcohol use should also be included to adjust.

Author Response

Thank you very much for reviewing our manuscript and commenting positively.

  1. Totally agree, as this manuscript was asigned to the endocrinology section of the journal, more information about these most commonly used stroke outcome scales in clinical care and stroke studies were provided.
  2. You are right that the additional predictors you mentioned would be interesting to be included in the multiple regression analysis. Nevertheless this study focused on the predictive value of glycemic variables and considering the sample size no more controlling co-predictos than the two known strongest ones (age and severetiy of stroke (NIHSS score)) could a priori be included in the prediction model for reasons of validity. We stated this point in our limitation section and stated that bigger patient collectives need to be examined for exhaustive prediction models.

Best regards

Reviewer 2 Report

Skin autofluorescence (SAF) reflects the in-patient load of so called advanced glycation end 10 products (AGEs) beyond HbA1c and represents a valid and quickly accessible marker of chronic 11 hyperglycemia. Authors analyzed  the predictive potential of SAF in comparison to HbA1c and acute 12 hyperglycemia on the functional outcome at 90 days after ischemic stroke in a cohort of patients 13 with DM. Authors  prospectively included 113 patients with DM type 2 hospitalized for acute 14 ischemic stroke. SAF was measured on the patient’s forearm by a mobile AGE-Reader mu© in arbi- 15 trary units. HbA1c and the area under the curve (AUC) of the blood sugar profile after admission 16 were assessed. Functional outcome was assessed via phone interview after 90 days. A poor outcome 17 was defined as a deterioration to a modified Rankin Scale score ≥3. A good outcome respectively 18 was defined as a modified Rankin Scale score <3 or as no deterorioration from premorbid level. 1 Patients with a poor outcome presented with higher values of SAF  20 than patients with a good outcome , but did not differ in HbA1c and 21 acute glycemia. In logistic regression analysis, age (significantly predicted a poor outcome, while HbA1c and acute glycemia 23 did not. Authors concluded that  SAF offers an insight into glycemic 26 memory and appears to be a significant predictor of poor stroke outcome in patients with DM exceeding HbA1c and acute glycemia. Thus, measuring SAF could be useful to identify specifically vulnerable patients on high risk for complications and poor outcome.

This is a very interesting and well conducted study

I have only minor comment to do : 

General: 

Did the authors evaluate the role of stroke subtype in the relationship between stroke outcome and mean SAF value; 

Discussion : 

Authors should add  a sentence about the role of inflammation in stroke outcome underlying possible relationship between inflammatory markers and SAF and they should add these citations on their reference section: 

Albanese A, Tuttolomondo A, Anile C, Sabatino G, Pompucci A, Pinto A, Licata G, Mangiola A. Spontaneous chronic subdural hematomas in young adults with a deficiency in coagulation factor XIII. Report of three cases. J Neurosurg. 2005 Jun;102(6):1130-2. doi: 10.3171/jns.2005.102.6.1130. PMID: 16028774; 
Della Corte V, Tuttolomondo A, Pecoraro R, Di Raimondo D, Vassallo V, Pinto A. Inflammation, Endothelial Dysfunction and Arterial Stiffness as Therapeutic Targets in Cardiovascular Medicine. Curr Pharm Des. 2016;22(30):4658-4668; Tuttolomondo A, Pedone C, Pinto A, Di Raimondo D, Fernandez P, Di Sciacca R, Licata G; Gruppo Italiano di Farmacoepidemiologia dell'Anziano (GIFA) researchers. Predictors of outcome in acute ischemic cerebrovascular syndromes: The GIFA study. Int J Cardiol. 2008 Apr 25;125(3):391-6. 

Author Response

Thank you very much for reviewing our manuscript and commenting positively.

  1. Yes, we investigated for different stroke etiologies acoording to TOAST (see baseline table) but without stastistic impact on stroke outcome. Cardioembolic strokes were the most frequent subtype though.
  2. Thank you for this imput, we highlighted the role of inflammation in ischemic damage and added the reference of inflammation and endothelial dysfunction (Della Corte V, Tuttolomondo A, Pecoraro R, Di Raimondo D, Vassallo V, Pinto A. Inflammation, Endothelial Dysfunction and Arterial Stiffness as Therapeutic Targets in Cardiovascular Medicine. Current pharmaceutical design (2016) 22:4658–68. doi:10.2174/1381612822666160510124801.). We also added and discussed another interesting reference from the same group on the role of inflammation in stroke and especially it’s potential systemic effects (Maida CD, Norrito RL, Daidone M, Tuttolomondo A, Pinto A. Neuroinflammatory Mechanisms in Ischemic Stroke: Focus on Cardioembolic Stroke, Background, and Therapeutic Approaches. International Journal of Molecular Sciences (2020) 21. doi:10.3390/ijms21186454.)

Best regards